# Turning a native or corroded Mg alloy surface into an anti-corrosion coating in excited $CO_2$

Yuecun Wang[1,2], Boyu Liu[1], Xin'ai Zhao[1], Xionghu Zhang[3], Yucong Miao[1], Nan Yang[1], Bo Yang[1], Liqiang Zhang[3], Wenjun Kuang[1], Ju Li [1,4], Evan Ma[1,5] & Zhiwei Shan [1]

Despite their energy-efficient merits as promising light-weight structural materials, magnesium (Mg) based alloys suffer from inadequate corrosion resistance. One primary reason is that the native surface film on Mg formed in air mainly consists of $Mg(OH)_2$ and MgO, which is porous and unprotective, especially in humid environments. Here, we demonstrate an environmentally benign method to grow a protective film on the surface of Mg/Mg alloy samples at room temperature, via a direct reaction of already-existing surface film with excited $CO_2$. Moreover, for samples that have been corroded obviously on surface, the corrosion products can be converted directly to create a new protective surface. Mechanical tests show that compared with untreated samples, the protective layer can elevate the yield stress, suppress plastic instability and prolong compressive strains without peeling off from the metal surface. This environmentally friendly surface treatment method is promising to protect Mg alloys, including those already-corroded on the surface.

[1] Center for Advancing Materials Performance from the Nanoscale (CAMP-Nano) & Hysitron Applied Research Center in China (HARCC), State Key Laboratory for Mechanical Behavior of Materials, Xi'an Jiaotong University, Xi'an 710049, P. R. China. [2] Shaanxi Key Laboratory of Low Metamorphic Coal Clean Utilization, School of Chemistry and Chemical Engineering, Yulin University, Yulin 719000, P. R. China. [3] Department of Materials Science and Engineering, State Key Laboratory of Heavy Oil Processing, China University of Petroleum, Beijing Changping 102249, P. R. China. [4] Department of Nuclear Science and Engineering, and Department of Materials Science and Engineering, Massachusetts Institute of Technology, Cambridge, MA 02139, USA. [5] Department of Materials Science and Engineering, Johns Hopkins University, Baltimore, MD 21218, USA. Correspondence and requests for materials should be addressed to L.Z. (email: lqzhang@cup.edu.cn) or to E.M. (email: ema@jhu.edu) or to Z.S. (email: zwshan@mail.xjtu.edu.cn)

Because of their lightweight, energy-efficient and environmentally friendly characteristics, magnesium (Mg) based alloys are desirable in applications ranging from transportation, 3C products and air/space industry to medical biomaterials, etc.[1–5]. However, compared with the passive films that form on aluminum or titanium, the native surface film on Mg formed upon exposure to air consists of mainly MgO and $Mg(OH)_2$, which is usually porous and unprotective, especially in humid environment[6–9]. Consequently, the applicability of Mg alloys has been severely compromised because of their high susceptibility to corrosion. One widely applied strategy in industry to protect Mg based materials from corrosion is to create a surface coating, as a barrier to isolate Mg metal from the external environmental attack[10,11]. The commonly used surface protection methods include chemical conversion coating, electrochemical plating, organic coating, anodic oxidation, thermal spray coating, etc.[10,12]. Unfortunately, the aforementioned treatments inevitably produce noticeable chemical wastes, going against the incentive of using Mg alloys as a green material. Moreover, the surface has to be pretreated carefully for the removal of pre-existing corrosion products so as to ensure the uniform coating coverage and adequate adhesion between the substrate and the coating[10]. The other strategy to mitigate corrosion of Mg-based materials is to design new Mg alloys with self-passivation ability. Most recently it was reported that the surface of a Mg–Li alloy can form a continuous and compact layer of lithium carbonate, via reaction of lithium oxide with $CO_2$ in air[13]. However, the protective $Li_2CO_3$ only forms when the alloy has a high Li content. In addition, a series of extrusion, heat treatment and rolling processes are needed to achieve the desired surface chemistry and microstructure. As such, this method is applicable only to a limited group of Mg alloys.

Considering the huge demands for light-weight structural materials, it is urgent to find an environmentally benign surface treatment method for Mg-based materials. To this end, one desirable way is to transform magnesium's native film formed in air into a protective scale directly. According to the classic Pilling–Bedworth (PB) rule, if the PB ratio (defined as the ratio of the molar volume of a metal oxide to the molar volume of the corresponding metal) is less than 1, the oxide under tensile stress is inclined to crack and provides no protective effect on the metal substrate (for example, the PB ratio of MgO is 0.81)[14,15]. When the ratio is larger than 1, the formed oxide is compressive and tends to be compact and protective. Thus, it is plausible to protect Mg alloys effectively by turning the indigenous surface oxide into other magnesium compounds with the PB ratio larger than 1.

In this work, we choose $MgCO_3$ because it is very stable in nature[16] (e.g., it has an ultra-low solubility of 0.02 g per 100 mL in water at ambient temperature and pressure, and can remain intact in water for quite a long time) and sufficiently compact (by assuming that $MgCO_3$ instead of MgO grows on Mg metal surface directly, the PB ratio of $MgCO_3$ is calculated to be 2.04). More importantly, it was reported that the protective carbonates/hydroxy carbonates-containing layer on the surface of Mg alloys can suppress corrosion by blocking the anodic and/or cathodic sites[17,18]. In principle, $MgCO_3$ can be obtained via the chemical reaction, $MgO + CO_2 \rightarrow MgCO_3$. At ambient temperature and pressure, the Gibbs free energy change of this chemical reaction is a negative value of $-21.4$ kJ $mol^{-1}$, i.e., the process is thermodynamically preferable even at room temperature (RT)[19]. But kinetically, at atmospheric pressure for the reaction of MgO with $CO_2$ to occur, heating to at least 400 °C[20] is required, a temperature high enough to deteriorate the properties of Mg alloys[10]. Therefore, it is essential to find a way to trigger the reaction at a temperature low enough to secure the integrity of Mg and its alloys. Here, we develop an easy approach for the carbonation of the air-formed oxide/hydroxide film or hydrate corrosion products on Mg alloys into a smooth, compact $MgCO_3$ protective surface layer via reaction with the $CO_2$ species activated by either high energy electron beam or glow discharge at room temperature. No extra need of heating or pretreatments makes this method promising to protect those small-sized or complex-shaped Mg alloy workpieces and to replenish those already-corroded but still functional components. Rather than having to mechanically clean away prior corrosion product, one can convert it directly to create new protective surface layer. The entire surface treatment process has been recorded in real time inside an environmental transmission electron microscope (E-TEM). Subsequent water immersion tests unambiguously prove the superior corrosion resistance of the $MgCO_3$ scale. In situ mechanical tests show that different from its brittle bulk counterpart, the thin $MgCO_3$ scale obtained has adequate deformability as well as excellent adhesion to the substrate.

## Results

**Reaction of MgO with excited $CO_2^*$ at room temperature**. It was reported that the electron beam (e-beam) can enhance the reaction rate of gas molecules with solids significantly via excitation of the gas species adsorbed on the solid reactant surface[21] or the electron-induced electrostatic field effect with memory effect for prior e-beam irradiation[22]. Inspired by these, we propose that by using e-beam illumination inside an environmental TEM to activate the adsorbed $CO_2$ (e.g., to turn them into ionic fragments of $CO_2^+$, $CO_2^{2+}$, $C^+$, $O^+$, etc.[23]; these $CO_2$ species with high chemical activity are denoted by $CO_2^*$) around the Mg sample, one may trigger the chemical reaction, i.e., $MgO + CO_2^* \rightarrow MgCO_3$, at room temperature. The schematic design is illustrated in Fig. 1.

To verify the feasibility of using electron beam-excited $CO_2^*$ to trigger the reaction, we first tested pure MgO (synthesized with the hydrothermal method, see more details about the processing in Methods section). Figure 2a shows the aggregated flakes of nanoscale MgO lamella, each being a single crystal with hexagonal morphology (see Supplementary Fig. 1). The aggregated MgO flakes remained stable after 14 min e-beam irradiation (Fig. 2b, the e-beam intensity was ~0.1 A $cm^{-2}$) in vacuum. Then 2 Pa $CO_2$ gas (99.99% purity) was flown into the chamber. Within a minute, the facets of the crystalline MgO started to curve. Thereafter, the flakes began to shrink, and the ensemble subsequently rolled into one single piece of reaction product (Fig. 2c). During this process, many highly mobile $CO_2$ bubbles were enclosed in the product before the gas could escape. The as-grown product was amorphous, which was later crystallized into $MgCO_3$ nanocrystals via heating (to 400 °C, using a Protochips in situ TEM heating device), as evidenced by the corresponding diffraction pattern in Fig. 2d. The entire in situ reaction process was recorded in Supplementary Movie 1. To further confirm that the excited $CO_2^*$ instead of other factors (e.g. e-beam heating, the creation of defect sites upon e-beam irradiation[24] or the electrostatic field effect within the oxide film[22]) is critical for the reaction, the following experiments were also performed. We blocked the e-beam and heated the MgO samples from RT to 300 °C under 2 Pa $CO_2$ atmosphere. The MgO flakes showed no visible change after holding at 300 °C for 15 min. While the temperature was down to RT, the MgO flakes reacted with $CO_2^*$ immediately upon exposure to e-beam irradiation (see Supplementary Fig. 2 and Supplementary Movie 2). The ongoing reaction of MgO with excited $CO_2^*$ stopped as soon as the e-beam irradiation was discontinued (see Supplementary Fig. 3), which suggests that this reaction shows no obvious memory effect[22] for the prior electron beam irradiation. This phenomenon cannot be

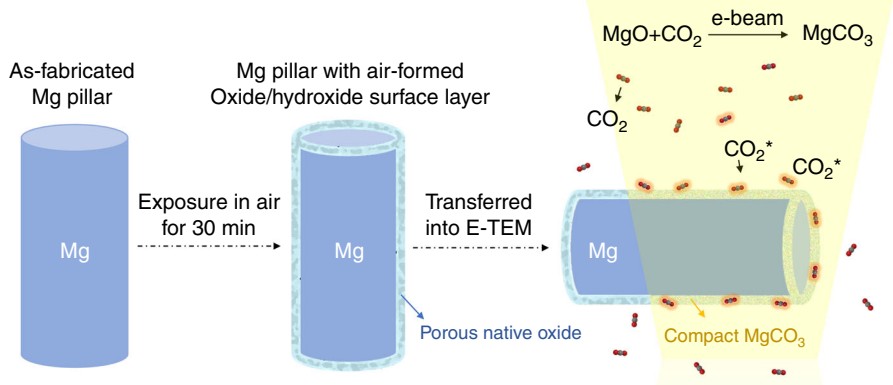

**Fig. 1** Schematic diagram showing how to transform the native surface to $MgCO_3$. A Mg micropillar with electron beam transparent size is fabricated by focused ion beam (left), and then exposures in air for 30 min to form a native oxide film (middle). After being transferred into the E-TEM, the air-formed surface layer is expected to react with the e-beam excited $CO_2^*$, and produce a compact $MgCO_3$ protective layer on the pillar's surface without any extra heating (right)

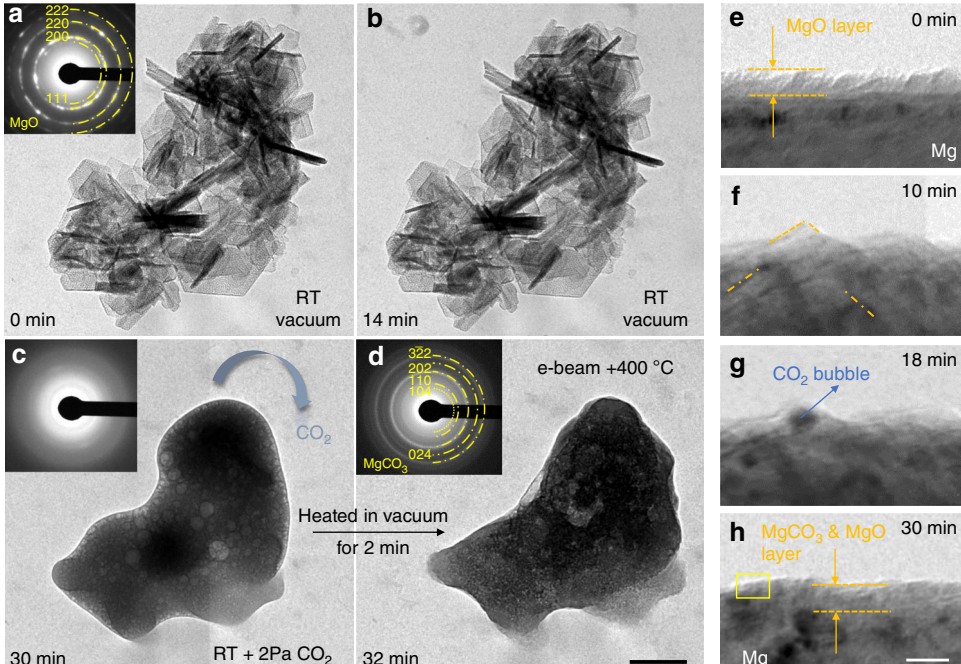

**Fig. 2** Validation of the proposed operation principle. **a** TEM image of the MgO crystal flakers synthesized by hydrothermal method. **b** The MgO flakers did not change after being exposed to a 300 keV, 0.1 A/cm² e-beam irradiation for 14 min. **c** The introduction of 2 Pa $CO_2$ atmosphere leads the reaction, i.e. $MgO + CO_2 \rightarrow MgCO_3$, occurred violently. The final product is that the MgO flakers were transformed into a piece of amorphous product with many bubbles in it. **d** After being heated to 400 °C in vacuum for 2 min, the amorphous product was crystallized. The inset selected area electron diffraction (SAED) pattern shows that it is nanocrystalline $MgCO_3$. **e–h** Surface evolution of a FIB-fabricated Mg pillar during the exposure in 2 Pa $CO_2$ and e-beam irradiation. TEM images showing transformation process of the native oxide layer to the compact nanocrystalline $MgCO_3$&MgO layer. Scale bars: **d** 200 nm, **h** 10 nm

rationalized by the e-beam irradiation-induced defect sites[24] or the electrostatic field effect[22] within the oxide film. Therefore, it is reasonable to speculate that reaction of MgO with $CO_2$ at RT should be attributed to the electron beam-stimulated excitation of $CO_2$ species adsorbed on the Mg oxide surface.

**Formation of $MgCO_3$ layer on Mg pillars' surface**. Next, we demonstrate that a $MgCO_3$ layer can be indeed formed on the surface of Mg/Mg alloys. Mg samples were purposefully fabricated into cylindrical micropillars to facilitate the real-time

observation and imaging in E-TEM (see Fig. 1). Nominally pure Mg pillars with electron transparent dimensions were fabricated using focused ion beam (FIB). The detailed fabrication process is described in the Methods section. After exposure in air for 30 min, these pillars were subsequently transferred into E-TEM. As can be seen in Fig. 2e, the air-formed layer has a thickness of about 8 nm. It appears continuous and dense because the thickness is thin enough to sustain the lattice mismatch. Then we treated the FIBed Mg pillars in 2 Pa $CO_2$ atmosphere with an e-beam intensity of ~0.1 A cm⁻². After about 10 min, some small crystalline facets (marked by orange dash lines) appeared on the

surface (Fig. 2f). Then the contrast of these facets diminished, and some nano-blisters began to form and grow (Fig. 2g). After 18 min of $CO_2$ exposure, the blisters stopped growing in size, and the blistered region gradually flattened out. The final surface layer after the reaction appeared more compact than the initial oxide/hydroxide layer, and its interface contrast with the metal substrate diminished, as shown in Fig. 2h. The final $MgCO_3$ layer is crystalline, as illustrated in the high resolution TEM image (Supplementary Fig. 4), which is magnified view of the framed zone in Fig. 2h. Electron energy loss spectroscopy (EELS) analysis was performed to identify the composition of the final reaction product (hereafter termed as scale). An obvious C-K edge appeared after the reaction (Supplementary Fig. 5a), which indicates the existence of carbon in the scale. Selected area electron diffraction (SAED) and high resolution TEM analysis confirmed the presence of both crystalline $MgCO_3$ and MgO after the reaction (Supplementary Fig. 5b, Supplementary Fig. 6).

The aforementioned phenomena can be rationalized as follows. Firstly, the molecular $CO_2$ was adsorbed on the native oxide surface, and then they were excited by e-beam forming $CO_2$ species ($CO_2^*$). The $CO_2^*$ diffused into and reacted with the MgO &$Mg(OH)_2$ layer to produce nanocrystalline $MgCO_3$, concurrently with the crystallization of some of the amorphous MgO and dehydration of $Mg(OH)_2$ under e-beam irradiation[6]. The excited $CO_2^*$ may also react with the remained $Mg(OH)_2$ via $Mg(OH)_2(s) + CO_2^* = MgCO_3(s) + H_2O(g)$ at the same time. It should be noted that along with the reaction, the newly formed $MgCO_3$ was dense and compact enough to stop the escape of previously entered $CO_2$ gas, which accumulated to form the surface blisters. When the surface layer was transformed into compact $MgCO_3$ completely, the reaction stopped because the adsorbed $CO_2^*$ on the surface of pillars couldn't react with the inner oxide further. However, the enclosed $CO_2^*$ would be consumed continuously, along with the disappearance of the gas-filled blisters. Those newly formed $MgCO_3$ and the residual oxide constitute the surface layer.

It is worth noting that compared with the more violent reaction of excited $CO_2^*$ with free standing MgO (Fig. 2b–d and Supplementary Fig. 7, separated oxide shell from the pillar) which produces amorphous products, the reaction with the MgO layer on Mg metal is more moderate and the final product is crystalline. Presumably, this is because the reaction rate is dependent on the specific surface area of reactants. Large specific surface area favors the adsorption of excited $CO_2^*$ species which in turn can accelerate the reaction process. Thermodynamically speaking, higher reaction rate tends to produce amorphous phase due to the highly nonequilibrium reaction conditions, while slower reaction rate is more conducive to the formation of crystalline $MgCO_3$ because the reaction condition is closer to equilibrium.

**Corrosion inhibiting effect of $MgCO_3$ protective film**. Immersion tests in aqueous environment were performed to examine the corrosion resistance of the as-grown $MgCO_3$ films. Usually, 3.5 wt% NaCl solution is the first choice for the corrosion medium. However, in order to decrease the microfabrication time, the thickness of the Mg material where the pillars were FIB-fabricated was usually less than 5 μm. We found that the thin area would dissolve in the 3.5 wt% NaCl solution within a period as short as 10 s. Given this, we chose deionized water as the corrosion medium, and mainly focused on the difference between the as-FIBed Mg pillars and those treated in $CO_2$. After immersion for 3 min, the as-FIBed pure Mg pillars suffered severe corrosion, while those treated in $CO_2$ remained intact (Fig. 3a). It clearly demonstrates that the $MgCO_3$ scale is effective in protecting the Mg metal. The same was found to be true for Mg alloys, using ZK 60 Mg alloy with small precipitates as an example (Fig. 3b). The

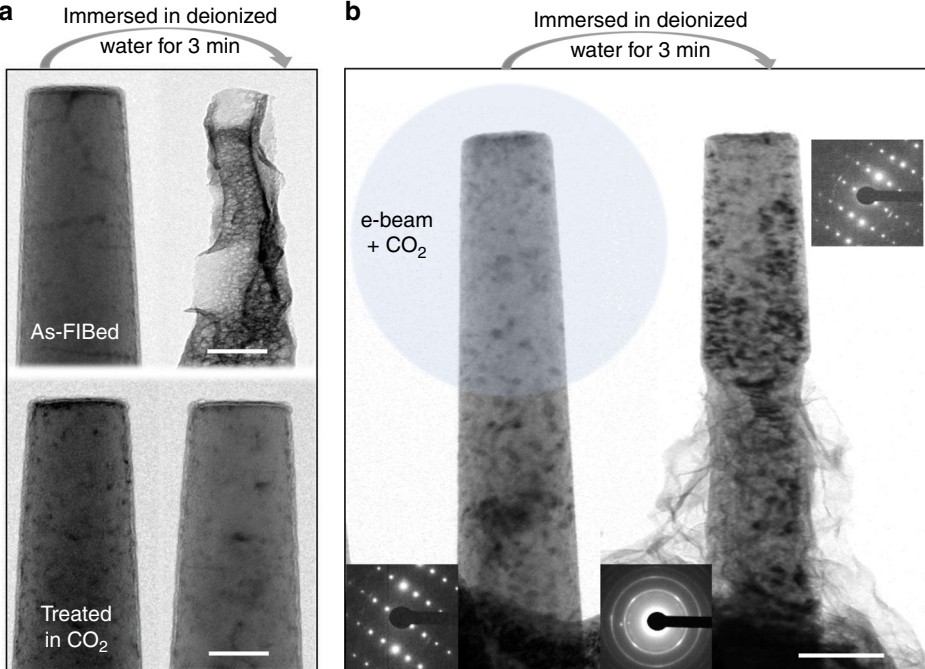

**Fig. 3** Turning native Mg alloy surface into protective coating. **a** TEM images of two Mg pillars (as-FIBed Mg pillar and the pillar treated in $CO_2$) before and after immersion in deionized water for 3 min. **b** Only half of the ZK 60 Mg alloy pillar was irradiated by e-beam in $CO_2$. The treated top half part was well protected from the aqueous environment and remained intact, as evidenced by the inset diffraction pattern, while the lower half was corroded seriously. Halos in the inset diffraction pattern indicate amorphous corrosion product. Scale bars: **a** 100 nm, **b** 200 nm

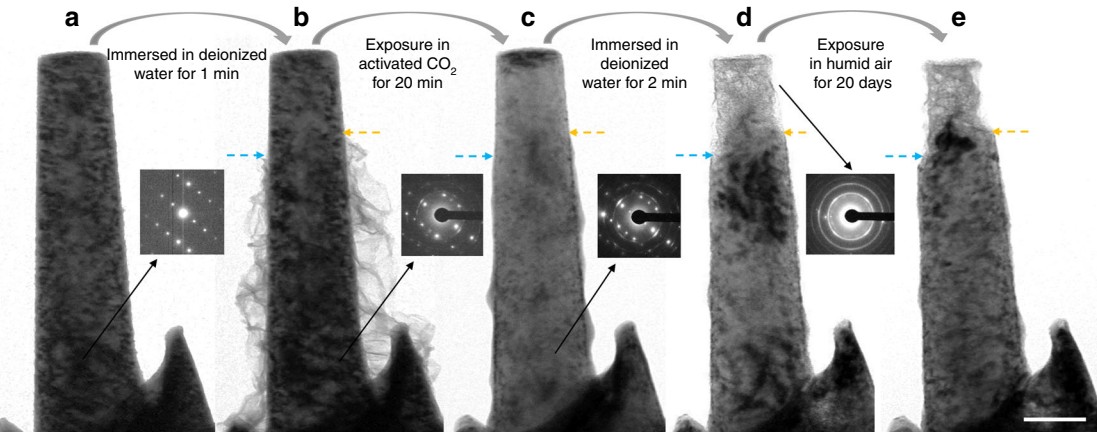

**Fig. 4** Turning corroded Mg alloy surface into protective coating. **a** As-FIBed ZK 60 Mg alloy pillar and the corresponding selected area electron diffraction (SADP). **b** After immersion in deionized water for 1 min, some fluffy $Mg(OH)_2$ &(MgO) corrosion product grew on the pillar's surface. Note the top part was bare. **c** After the corroded pillar was exposed in $CO_2$ and the e-beam irradiation atmosphere for 20 min (only the part with fluffy corrosion product was irradiated by e-beam), the newly formed $MgCO_3$ covered the bottom part of the pillar. **d** After water immersion, only the top part of the pillar was corroded. The region covered with $MgCO_3$ remained unchanged. **e** The top part was further corroded, and the protected part remained intact after exposure in humid air (relative humidity: 65–80%, temperature:18–26 °C) for 20 days. Scale bar, 200 nm

e-beam was converged onto the top half part of a ZK 60 pillar only. After immersion in deionized water for 3 min, an obvious boundary between the two parts can be seen in the TEM image: the treated top half part was well protected from the aqueous environment, while the bottom half part was corroded seriously. This demonstrated directly that it is the $CO_2$ adsorbed on the porous native oxide surface layer instead of the free moving $CO_2$ gas molecules that is activated by e-beam. At the same time, it can be concluded that the $MgCO_3$ film is an effective barrier to keep Mg/Mg alloys from aqueous environmental attack.

**Turing corrosion products to $MgCO_3$ protective layer**. The discussions above deal with Mg (and ZK 60) having only its native oxide on the surface. The corrosion products of Mg alloys under humid air or aqueous conditions are typically a mixture of MgO and $Mg(OH)_2$[6,25,26]. Since the excited $CO_2^*$ can react with MgO at RT, it should be able to react with the corrosion products on Mg alloy to form $MgCO_3$ as well. To confirm this, we pre-corroded some Mg alloy pillars. The as-FIBed Mg alloy pillar (Fig. 4a, Supplementary Fig. 8) was immersed in deionized water for 1 min, and then dried in air for 1 h. The immersion time was controlled to yield enough corrosion product on the surface while avoiding extensive corrosion into the interior. For better comparison, we intentionally chose a long pillar (Fig. 4b) that was not corroded completely, i.e., the entire pillar, except for its free end part, was covered by fluffy $Mg(OH)_2$ (MgO). The corrosion probably started from the pillars' base part, which can be regarded as a water reservoir, and then the corrosion process crept up along the pillar. The inset diffraction pattern from the lower part of the pillar indicates that the metal substrate remains single crystalline despite of its corroded surface. The lower part with fluffy corrosion products was subsequently exposed to e-beam illumination in 4 Pa $CO_2$ for 20 min. The fluffy film shrank around the pillar due to reaction with $CO_2^*$. The reaction product wrapped around the pillar's lower part, resulting in a smooth surface (Fig. 4c). More details are shown in Supplementary Movie 3. Figure 4d shows this sample after immersion in deionized water for two minutes. The bare top part was heavily corroded, while the $MgCO_3$-protected part was intact. Further exposure in humid air (relative humidity: 65–80%, temperature:18–26 °C) for 20 days also confirmed the protection effect of the $MgCO_3$ scale (Fig. 4e). As such, even when there are

significant pre-existing corrosion products on the surface, they would react with the excited $CO_2^*$ to produce a compact $MgCO_3$ scale that is protective against further corrosion. In other words, rather than having to mechanically clean away the prior corrosion products, one can convert them directly to a new protective surface layer. The thickness of the $MgCO_3$ scale can be tuned by adjusting the corrosion time of Mg and its alloys. Besides protecting the Mg alloys from environmental attack, the $MgCO_3$ film can effectively decrease the oxidation rate of Mg at elevated temperatures (Supplementary Fig. 9).

**Mechanical tests**. A ceramic surface coating from anodizing or spraying is usually quite brittle[10], which could degrade the mechanical performance of Mg alloys. Therefore, it is important to gauge how much change in mechanical behaviors would result from the $MgCO_3$ scale. Surprisingly, the formed protective layer not only has very good adhesion to its substrate during the plastic deformation but also noticeably improves the mechanical properties of the small Mg samples, in terms of yield stress and compressive strain to failure, as shown in Fig. 5. The engineering stress-strain curves from the compression tests of two typical samples are shown in Fig. 5a. The yield stress of the treated pillar (black) is much higher than that of the as-fabricated (red) Mg pillar.

The morphologies before and after compression (Fig. 5b) suggest that the $MgCO_3$ protective layer not only delays plastic instability, but also has excellent adhesion to its substrate during the plastic deformation. Presumably, this is because the gliding dislocations are blocked by the strong but ductile $MgCO_3$ film, and their accumulations or interactions lead to the high yield/flow stress, strain hardening and uniform plastic flow. Figure 5c compares the yield stress and compressive strain of the two kinds of pillars. Clearly, the yield stress of the treated pillars is almost twice of those untreated. The strain to failure (judged by the generation of the large shear offset) of the as-FIBed Mg pillars is quite low (2–4%). Interestingly, for the treated pillars, we gradually increased the compression amount, and found that they deformed with smooth plastic flow and the protective scale has not spalled even when the maximum plastic strain reached ~23%, suggesting adequate adhesion with the Mg substrate. Presumably, this is because the $MgCO_3$ scale is thin enough to flow, with a compact initial structure without significant flaws.

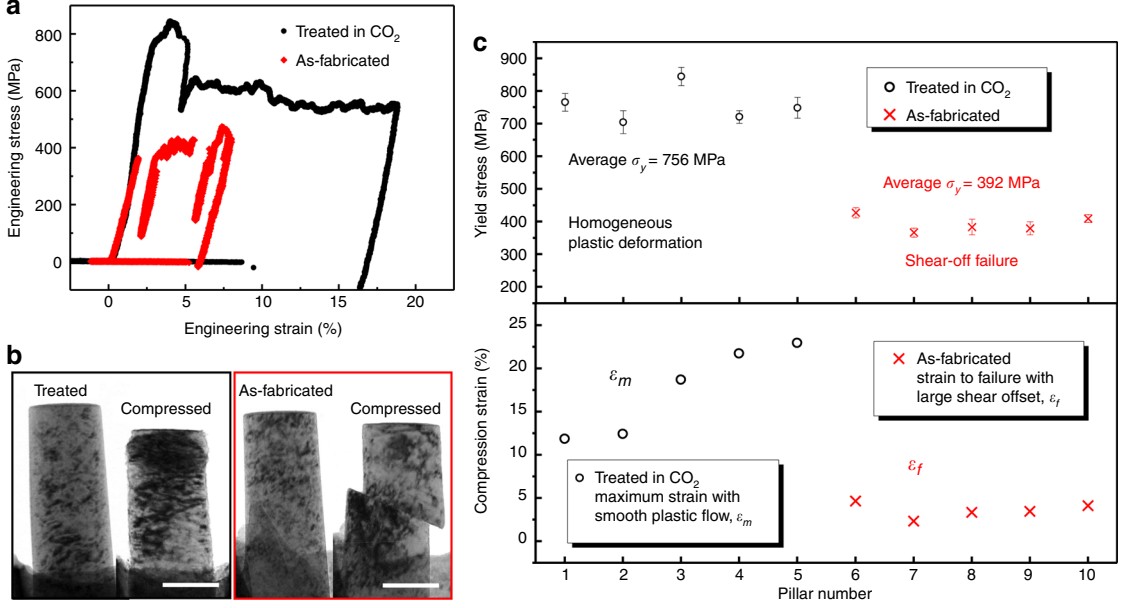

**Fig. 5** Effect of the treated surface layer on the mechanical properties of the samples. **a** Typical compressive engineering stress-strain curves of an as-fabricated Mg pillar (red) and a pillar treated in $CO_2$ (black). Note both the yield stress and plastic deformation ability are improved significantly. **b** Bright-field TEM images of the Mg pillar treated in $CO_2$ (framed in black) and an as-fabricated Mg pillar (framed in red) before and after compression. The $MgCO_3$ protective layer showed great adhesion to its substrate and excellent compatible deformation capability. **c** Statistics of the yield stress and compressive strain vs. the Mg pillars. Different from those as-fabricated Mg pillars that fail with large shear offset, the pillars with $MgCO_3$ protective layer deform homogeneously with smooth plastic flow with the maximus compressive strain up to 22%. The error bars are two standard deviations. Scale bars, 200 nm

There should be a critical thickness, above which the $MgCO_3$ scale would behave similarly to its brittle bulk counterpart. See more data in Supplementary Fig. 10.

## Discussion

The carbonation reaction and the protective effects demonstrated so far were for submicron-sized Mg and Mg alloys. It is necessary to scale up this method for protecting the bulk Mg alloys. We note that the main effect of e-beam in the E-TEM experiments is to produce excited $CO_2^*$ species in plasma-like state and promote their chemical conversion. Plasma usually contains ionized and excited states of atoms, molecules and radicals normally not present at ambient temperature, and these excited species possess high reactivity that is otherwise only seen at high temperatures (>1000 K)[27]. It thus follows that in lieu of e-beam irradiation, for bulk samples one may be able to use $CO_2$ plasma to produce a similar outcome as in E-TEM experiments. Non-thermal plasma is in fact a mature technology in industry and it can be produced by dielectric barrier discharges[28], glow discharge[29], etc. To test this possibility, we put MgO powders in glow discharge induced the $CO_2$ plasma (see more details in Methods). MgO powders indeed reacted with the $CO_2$ plasma, producing $MgCO_3$ (Supplementary Fig. 11, Supplementary Fig. 12) just like that obtained in E-TEM experiments (Fig. 2). Therefore, it is feasible to grow $MgCO_3$ protective layer on bulk Mg alloys in $CO_2$ plasma. It should be noted that the reaction mechanism of MgO with e-beam excited $CO_2$ is fundamentally different with $CO_2$ plasma. The glow discharge activated $CO_2$ molecules in the gas phase directly form the $CO_2$ plasma with ionic fragments and radicals of $CO_2^*$. Due to the fast motion of gas species, the formation of $MgCO_3$ occurs in any areas that can be reached by the $CO_2$ plasma. However, the situation is different for the high energy e-beam irradiation. Compared to the $CO_2$ species adsorbed on the native oxide surface, the ionization rate of those free moving $CO_2$ gas molecules is much lower[23]. Therefore, $MgCO_3$ forms only in

the local areas with absorbed $CO_2$ species and being exposed to the e-beam irradiation.

Considering that the native film on Mg surface is only tens of nanometers, the formed $MgCO_3$ protective layer is too thin to protect the bulk Mg. Inspired by the reaction of excited $CO_2$ with the corrosion product on the surface of Mg micropillars, we intentionally immersed the bulk Mg samples in deionized water for 24 h to form a plate-like pre-corroded layer with a micron-scale thickness. After immersion, the specimens were treated in $CO_2$ plasma for some time and the corroded surface with plate-like product were carbonated resulting in a relatively flat and dense $MgCO_3$ surface layer (Supplementary Fig. 13, Supplementary Note 1). Then the inhibition effects of the $MgCO_3$ films were evaluated in 3.5 wt.% NaCl solution using potentiodynamic polarization technique. Compared with their counterparts, the corrosion resistance of bulk Mg samples with $MgCO_3$ protective film have been obviously enhanced: the corrosion potential ($E_{corr}$ vs.SCE) increases from −1.58 to −0.73 V and the corrosion current density ($i_{corr}$) decreases by two orders of magnitude.

After corrosion tests, large corrosion craters can be clearly observed even by naked eyes on pristine Mg surface, while the surfaces of the samples treated in $CO_2$ for 2 h are still intact and there are no visible cracks on the SEM images. See Supplementary Fig. 13 and Supplementary Table 1 for details. The corrosion inhibition effects of the $MgCO_3$ protective film are better than the anodized coating[30], and can even be comparable with the composite coatings obtained by micro-arc oxidations[31,32]. Therefore, for bulk applications, the $MgCO_3$ layer should be optimized so that it is thick enough to maximize the corrosion inhibition effect while at the same time thin enough to ensure its deformation compatibility. This can be achieved by optimizing the treatment parameters used to generate the $MgCO_3$ layer.

Finally, we comment on examples that may potentially benefit from our new surface treatment technique. One case is for Mg alloy cardiovascular stents. The surface treatment method we

reported here offers an effective control that can tune the corrosion resistance and ensure their reliability after implantation. Meanwhile, the MgCO₃ layer is expected to leave behind no toxic substance and can be made sufficiently thin to ensure the elastic compliance and strength of the stents while offering adequate corrosion protection. The treatment may alleviate the current problem: the rapid surface corrosion rate of Mg alloys causes accumulation of corrosion products adjacent to the implant and increase in local pH of the body fluid, which in turn impedes their applications in biodegradable cardiovascular stents despite their outstanding biocompatibility, resorbability and mechanical properties[33,34]. As a second example, our treatment is especially suitable to generate a protective coating on micro-nano scaled Mg alloys devices. Mg alloys are very suitable for electronic components or precision instruments industry[35] because of their high heat resistance, good resistance to electromagnetic interference and good shock absorption ability. However, up to now, their poor corrosion resistance is a major hurdle that hampers their use in micro/nano-electro mechanical systems. Our experiments using e-beam or in non-thermal $CO_2$ plasma demonstrate an effective surface treatment for the micro- and nano- scale components, which cannot be achieved through any other corrosion protection methods.

Our findings reported here is a success story in making use of the state-of-art in situ environmental TEM to explore a pressing practical problem, from ideas to solution.

## Methods

**Synthesis of MgO powder via hydrothermal method**. Single crystal magnesium oxide flakes were synthesized using hydrothermal method. 0.01 mol $MgSO_4$ powder was added into a Teflon-lined autoclave of 100 ml capacity. Then the autoclave was filled with 0.1 mol·L⁻¹ ethylenediamine solution up to 70% of the total volume and kept at 180 °C for 20 h. After cooling to room temperature, the white powders were collected, washed with distilled water and ethanol for several times, and then dried in vacuum at 280 °C for 4 h. The products were subsequently calcined in a muffle furnace, first held at 280 °C for 1 h, then at 380 °C for 2 h, and finally at 450 °C for 2 h. The temperature was increased very slowly to preserve the nanocrystalline features of the final MgO products.

**Preparation of the Mg /Mg alloys micropillars**. We used bulk Mg single crystal (purity of 99.99% wt%), commercial ZK60 Mg alloy (6 wt% Zn, < 0.5 wt% Zr) and AZ31 (3 wt% Al, 1 wt% Zn, 0.2 wt% Mn) Mg alloy as starting material. Bulk samples are made into thin semicircle TEM specimens. At the center of the specimens, pillars with diameter ranging from 200 to 350 nm are micromachined using FIB (Helios NanoLab 600i dual-beam FIB system) under 30 kV acceleration voltage with Ga ion beam current sequentially decreasing from 440 pA (coarse cutting) to 1.5 pA (fine polishing).

**In situ experiments inside the environmental TEM**. The experiments were conducted in an environmental TEM (Hitachi H9500, 300 keV). The specimen chamber was evacuated to a base vacuum of 10⁻⁴ Pa. High-purity $CO_2$ (99.99%) was introduced through a needle valve into the specimen chamber. The pressure was controlled in the range of 2–4 Pa, measured by a Pirani vacuum gauge near the sample. The intensity of the electron beam illuminating the pillars was 0.02–0.2 A cm⁻². All the real-time observations were recorded using a Gatan 832 camera at 5 frames per second.

In situ compression test was conducted using a Hysitron PicoIndenter (PI 95 H1H) inside the E-TEM. The tests were all controlled under a fixed displacement rate of 2 nm·s⁻¹. For all the samples, engineering stress was calculated by dividing the load by cross section area $A$, which is calculated as $\pi d^2/4$, where $d$ is the nominal diameter measured at the half height of the pillars. The engineering strain was defined as the ratio of the deformation displacement of the pillar (i.e. the displacement reading minus the contribution from the substrate) to its initial height (the distance from top to the substrate).

**Reaction of CO₂ plasma with MgO powders**. Hydrothermal-synthesized MgO powders were dispersed on Si wafers (for the subsequent SEM characterization) and copper mesh grids (for the subsequent TEM characterization). They were put inside a lab-made glow discharge chamber. High-purity $CO_2$ (99.99%) with a flow rate of 100 sccm was introduced into the chamber through pipeline. The discharge voltage was 550 V, with a current of 50 mA and the electrode spacing around 1 cm. Treatment time was ~30 min.

**Potentiodynamic polarization tests**. The commercially pure (99.95 wt%) Mg samples were machined and polished with progressively finer SiC paper up to 1000 grit size. The samples were rinsed with distilled water and ethanol before immersion in deionized water. The $CO_2$ plasma treatment parameters of the Mg samples immersed in DI water are: high-purity $CO_2$ (99.99%) with the 100 sccm flow rate, the discharge voltage was 550 V, with a current of 50 mA and the electrode spacing around 1 cm. These samples were electrically connected with a copper wire using the conductive silver epoxy and then casted in epoxy resin with an exposed area of ~1 cm² for tests.

The potentiodynamic polarization curve tests were carried out on an electrochemical workstation (CHI 660E) with the conventional three-electrode system. Pt sheet and Ag/AgCl electrode or saturated calomel electrode (SCE) were used as counter electrode and reference electrode, respectively. In addition, the polarization curves of different working electrodes (pristine Mg, immersed Mg and plasma-treated Mg) were tested at a scanning rate of 1 mV s⁻¹ in 3.5 wt% NaCl solutions.

## Data availability

The data that support the findings of this study are available from the corresponding authors on request.

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

## Acknowledgements

The authors acknowledge the supports by the National Key Research and Development Program of China (No. 2017YFB0702001) and Natrual Science Foundation of China (51621063, 51601141, 51401239), Science and Technology Department of Shaanxi Province (2016KTZDGY-04-03 and 2016KTZDGY-04-04), China Postdoctoral Science Foundation (2016M600788), Science Foundation of China University of Petroleum, Beijing (Nos. 2462018BJC005, C201603). We also appreciate the support from the International Joint Laboratory for Micro/Nano Manufacturing and Measurement Technologies and the Collaborative Innovation Center of High-End Manufacturing Equipment and 111 project (B06025). J.L. acknowledges support by NSF ECCS-1610806. E.M. acknowledges support from U.S. DoE-BES-DMSE, under Contract No. DE-FG02-16ER46056.

## Author contributions

Z.W.S. conceived and designed the project. Y.C.W., X.A.Z. and Y.C.M. fabricated the samples and carried out the in situ E-TEM experiments. N.Y. and Y.C.W. collected the EELS data. L.Q.Z. synthesized the nanoscale MgO lamella and treated them in $CO_2$ plasma. X.H.Z. and L.Q.Z. performed the electrochemical tests. B.Y. treated the MgO lamella in $CO_2$ plasma and characterized them in TEM. Y.C.W., B.Y.L., E. M., Z.W.S., J.L. and W.J.K. wrote the paper. All the authors discussed the results and commented on the manuscript.

## Additional information

**Competing interests:** The authors declare no competing interests.

