## [Peer Review File · Nature Communications]

Reviewers' comments:

Reviewer #1 (Remarks to the Author):

The paper herein seeks to communicate anti-corrosion of Mg by excited CO₂.

The paper has several fundamental flaws, some factual (which is unsatisfactory) and others, contextual (oblivious to prior works with much more novelty and better performance).

The reader is disillusioned immediately by the abstract. Actually, the use of Mg has been steadily growing, rather successfully, over the past 15 years, in portable electronics, cast components in cars (casings, steering wheels), etc. As such, the first line in the abstract is simply not correct, and not supported.

The second line in the abstract, proves (as does the paper in many locations) that authors do not understand Mg corrosion, nor have they read the literature. The film on Mg is not Mg... its nominally Mg(OH)₂, often with some traces of MgO. It is a hydrated oxide that forms rapidly in (moist) air and water.

The reduction of Mg corrosion rate by CO₂ (to form carbonates) has been well and truly published many times before. Many industrial processes even use CO₂ as a corrosion inhibitor....

The claim that "This environment-friendly surface treatment method is expected to be universal to protect Mg-based materials, even those already-corroded Mg alloy workpieces." is entirely overstated. Cladding with aluminium can also do the same, so such a statement is not adding new science, or even impact. When is Mg used as a boldly exposed, uncoated material, such that when it corrodes, excited laser can help? Nowhere. Are you going to laser excite the inside of a cell phone? A wrapped steering wheel, or, a gearbox housing on a helicopter (especially if there is section loss). The answer is no.

The statements like "becoming a type of very promising and revolutionary material" for Mg are unfortunately, rather unsatisfactory. Whole aircraft were made of Mg back in the second world war era.... and thus, revolutionary is a gross overstatement.

A timeline of Mg (and corrosion), which incidentally corrects such overstatements, and incidentally shows that CO₂ is not novel, is given by Esmaily et al (Progress in Materials Science).

The reactions showing Mg going to MgCO₃ can't be accurate, as there is a lot of Mg(OH)₂ on the surface. As such, what happens to the Mg(OH)₂? is it gone?

The list of references is particularly bad I'm afraid. There have been excellent Mg surface film observations by Kish et al (all missed!).

All that Figure 3b shows is a galvanic couple, not protection at all. One side is the cathode, one the anode. This is how corrosion works.... All it takes is a small potential difference and this occurs. This is not so whopping.

The claim in Figure 4 that "Turning corroded Mg alloy surface into protective coating." is an overstatement. There is quite a lot of metal loss.

Irrespective of much of the above, the authors describe a coating... in essence. It is not a protective coating of value unless it is protective in a defect, or self forming (like a passive film). The Mg-Li film previously demonstrated on Mg-Li alloys is self healing (by self reforming), so it protects defect. Coating Mg with paint or plating with Cu or Ni is common. None of the latter protect defects, and neither does the coating described here. This means it is not actually interesting, or useful.

Reviewer #2 (Remarks to the Author):

This is an interesting paper about the use of room temperature pretreatments to improve the protective oxide of Mg. These authors demonstrate via in situ TEM (environmental and mechanical) that the native oxide on Mg can be dramatically improved by exposure to ionizing CO₂ gas (using electron beam) to change the native oxide to a protective MgCO₃ layer. The authors claim that this oxide is more resistant to corrosion and more adherent to the underlying Mg. The work is of high quality, especially the in situ microscopy. Sufficient details in the paper and the supplement are given for others to reproduce the data.

However, the corrosion experiments were done by only immersion in deionized water. John T. Yates, Jr. paper in Langmuir 2001, 17, 2146-2152 also shows that using different oxidation conditions can improve corrosion behavior as well, where they used an electrochemical cell to verify improved corrosion behavior. This paper could be used as an example of how to further characterize the MgO versus the MgCO₃. The authors could also demonstrate improved oxidation behavior as well, as adhesion of the oxide is important for high temperature corrosion field, where cyclic oxidation using TGA is a standard method to demonstrate improved oxidation behavior.

Another point is that more specifics of the SAED analysis could be given to be more convincing that the oxide formed is MgCO₃ (or mixed with MgO).

As the authors have access to an ETEM, could the authors expose the samples in situ to O₂ or H₂O vapor to determine if the MgCO₃ in contrast to the initial MgO does slow down or change oxidation behavior at higher temperatures?

In the conclusion section, it would also be beneficial to the general audience to reiterate the broad application of Mg in technology and what environments that Mg would be exposed to in these technological applications. It would also be helpful to again provide some thermodynamics or kinetics of this reaction (MgO + CO₂ → MgCO₃) in the conclusion section in order to give a broader implication of how to select the pre-treatments to create better protective oxides on other metallurgical systems.

Some minor points:

SM video 2: the beginning should be edited out.

Subtitles should be removed (e.g., Verification of Idea).

Some careful minor edits would improve clarity (e.g., Figure 2b "... MgO flakes kept unchanged...", would read better as "... MgO flakes do not change...").

To summarize, this is an interesting paper that should influence the field of corrosion, oxidation, and nanoscience and technology, as the authors clearly demonstrate that different pretreatments especially reactive gas species can improve properties significantly. Although the TEM is excellent and convincing, the claim that the corrosion properties are improved through exposure to deionized water could be better. It would benefit the paper to carry out the more standard electrochemical corrosion or higher temperature cyclic oxidation tests to demonstrate that this pretreatment improves Mg resistance to corrosion or oxidation properties.

Reviewer #3 (Remarks to the Author):

This paper demonstrated that magnesium oxide could be converted into magnesium carbonate by excited CO₂ in an environmental TEM, and the formed magnesium carbonate film could enhance the mechanical properties and inhibit the corrosion of magnesium in water. These are innovative findings. The most impressive experiment in this study was the in-situ observation of the formation process of magnesium carbonate. Therefore, the reviewer would like to recommend

publication of this paper.

However, this paper can be further improved if the following issue could be properly addressed. Corrosion resistance is critical for a surface film on magnesium. Since the formed magnesium carbonate film in this paper was claimed to have superior corrosion resistance in water, it would be worthwhile to compare this film with a simple coating, e.g. an anodized coating, on the magnesium under the same condition. The reviewer believes that such a comparison will interest many more scientists and engineers. If the magnesium carbonate film can prove to be more corrosion resistant than the simple coating, then this new film formation technique may open up a new way of coating magnesium alloys in practical applications.

Response to the reviewer #1:

Comment 1:

The paper herein seeks to communicate anti-corrosion of Mg by excited CO₂. The paper has several fundamental flaws, some factual (which is unsatisfactory) and others, contextual (oblivious to prior works with much more novelty and better performance).

Reply:

In the following, we respond point-to-point to each comment of the reviewer.

Comment 2:

The reader is disillusioned immediately by the abstract. Actually, the use of Mg has been steadily growing, rather successfully, over the past 15 years, in portable electronics, cast components in cars (casings, steering wheels), etc. As such, the first line in the abstract is simply not correct, and not supported.

The statements like "becoming a type of very promising and revolutionary material" for Mg are unfortunately, rather unsatisfactory. Whole aircraft were made of Mg back in the second world war era..... and thus, revolutionary is a gross overstatement.

Reply:

We have modified the first sentence in abstract to “Despite their energy-efficient merits as light-weight structural materials, magnesium (Mg) based alloys suffer from inadequate corrosion resistance.”. We also removed “very promising and revolutionary material” in the first sentence of Introduction.

Comment 3:

The second line in the abstract, proves (as does the paper in many locations) that authors do not understand Mg corrosion, nor have they read the literature. The film on

Mg is not MgO.... its nominally Mg(OH)₂, often with some traces of MgO. It is a hydrated oxide that forms rapidly in (moist) air and water.

Reply:

We thank the reviewer for pointing out this inadequate statement.

The second sentence in abstract is “One primary reason is that the native surface oxide on Mg formed upon exposure to air consists of mainly MgO, which is porous and unprotective, especially in the humid environment.” Initially, we intended to emphasize the first product **upon** exposure to air is mainly MgO. According to the XPS and TEM results of Nordlien *et al.* (Nordlien, J. H., *et al.* "A TEM investigation of naturally formed oxide films on pure magnesium." *Corrosion science*, 1997), the initial film formed immediately after exposing fresh surface by scratching in air contains 50%~60% wt% (40%~50% at%) magnesium hydroxide, and the other composition is MgO. Certainly, our claim is not entirely right in context. So, we revised this statement as “One primary reason is that the native surface film on Mg formed in air mainly consists of Mg(OH)₂ and MgO, which is porous and unprotective, especially in humid environment.” Elsewhere in the manuscript, we also replaced MgO with oxide/hydroxide or native layer formed in air.

In addition, it should be noted that Nordlien *et al.* also reported the water evaporation from the hydrated zones of films formed on Mg and subsequent crystallization of the dehydrated zones into MgO occurring in TEM analysis due to high energy electron irradiation. (Nordlien, Jan Halvor, *et al.* "Morphology and structure of oxide films formed on MgAl alloys by exposure to air and water." *Journal of the Electrochemical Society*, 1996. Nordlien, J. H., *et al.* "A TEM investigation of naturally formed oxide films on pure magnesium." *Corrosion science*, 1997). So, the carbonation of the air-formed film on our pillars' surface we observed in TEM should be dominated by the reaction $MgO + CO_2 = MgCO_3$.

Comment 4:

A timeline of Mg (and corrosion), which incidentally corrects such overstatements,

and incidentally shows that CO₂ is not novel, is given by Esmaily et al (Progress in Materials Science).

The reduction of Mg corrosion rate by CO₂ (to form carbonates) has been well and truly published many times before. Many industrial processes even use CO₂ as a corrosion inhibitor.....

Reply:

The reviewer thought that our work is not novel because the inhibitive effect of CO₂ on Mg corrosion has been reported. In the following, we will summarize the relevant researches progress and demonstrate their huge differences with our work.

It was reported that CO₂ inhibits Mg corrosion by 1) lowering pH (when dissolved in water, CO₂ forms carbonic acid), impeding the development of macroscopic corrosion cells and resulting in inhibition of pitting corrosion; 2) forming a magnesium hydroxy carbonate or carbonate-containing surface film, which can lower the conductivity of the electrolyte, block the anodic and cathodic reactions.

(Martell A E, Smith R M. *Critical stability constants[M]*. New York: Plenum Press, 1974.

R. Lindström, J.-E. Svensson, L.-G. Johansson, J. *Electrochem. Soc.* 149, 2002.

Lindström, Rakel, et al. "Corrosion of magnesium in humid air." *Corrosion Science*, 2004.

Shahabi-Navid, Mehrdad, et al. "NaCl-induced atmospheric corrosion of the MgAl alloy AM50-the influence of CO₂." *Journal of The Electrochemical Society*, 2014.

Esmaily, M., et al. "Fundamentals and Advances in Magnesium Alloy Corrosion." *Progress in Materials Science*, 2017.)

Based on these, we see that formation of Mg carbonate-containing film is only a partial reason for the inhibitive effect of CO₂ gas on the Mg corrosion. Moreover, the following two points should be noted:

- 1) The mechanism of CO₂ as corrosion inhibitor has not been fully understood. A study even shows that WE43 Mg alloy dissolved ambient levels of CO₂ slightly accelerated the corrosion rate and increased general surface corrosion compared to a CO₂-free environment: the average corrosion rate in the ambient CO₂ was 0.16 mg/cm²·day compared to 0.09mg/cm²·day in the absence of CO₂ (Kaminski, Daniel Thomas. *Corrosion Inhibition of Magnesium Alloys and*

Influence of Atmospheric Carbon Dioxide. Diss. The Ohio State University, 2016.) This cannot be well explained according to the present theory.

- 2) According to the previous papers, only in humid or aqueous environment with the appearance of CO₂ gas, the magnesium hydroxy carbonate or carbonate can be produced. In the usual atmospheric environment (about 400 ppm CO₂ gas), the surface film is a mixture of Mg oxide, hydroxide, hydroxy carbonate or carbonate and the first two are the majority. So, this film cannot be as compact as the pure MgCO₃ film we proposed. That is to say, the mixture film cannot stop the further corrosion of inner metal effectively. Actually, almost all Mg alloys in the atmospheric environment with CO₂ gas are still facing serious corrosion problem, that's why so many researchers never stop looking for different methods to further improve the corrosion resistance.

The protective effect of Mg carbonate mentioned in literature prompted us to find a new way to grow MgCO₃ protective film at room temperature, and this is the point we are making in this paper. Our novelty mainly lies in how to transform the native porous oxide/hydroxide into a uniform and compact MgCO₃ protective film on the uncorroded or slightly corroded Mg alloys' surface controllably in a dry environment without extra heating. The new environmental TEM technique, which is rarely used in the corrosion community, played a key role in our advances.

Comment 5:

When is Mg used as a boldy exposed, uncoated material, such that when it corrodes, excited laser can help? Nowhere. Are you going to laser excite the inside of a cell phone? A wrapped steering wheel, or, a gearbox housing on a helicopter (especially if there is section loss). The answer is no.

Reply:

Firstly, the CO₂ laser with CO₂ plasma are completely different and we did not mention "laser" in this manuscript. The CO₂ excited by electron beam or glow

discharge exists in the state of plasma and can react with MgO or Mg(OH)₂ at room temperature; this is what we want to demonstrate.

Secondly, the reviewer's worries are not necessary. If the devices are not sealed, the CO₂ plasma can diffuse onto the Mg alloys parts' surface to react with existing corrosion products or the Mg alloys parts may be disassembled from the device and then immersed in CO₂ plasma. If the devices are sealed, there are no worries about corrosion. And certainly, it is OK to pre-treat the Mg alloys in advance to protect them from corrosion.

Comment 6:

The claim that "This environment-friendly surface treatment method is expected to be universal to protect Mg-based materials, even those already-corroded Mg alloy workpieces." is entirely overstated. Cladding with aluminium can also do the same, so such a statement is not adding new science, or even impact.

Reply:

We disagree the statement "This environment-friendly surface treatment method is expected to be universal to protect Mg-based materials, even those already-corroded Mg alloy workpieces" is **entirely** overstated. It is a reasonable conclusion and expectation according to our experimental results. The reviewer's comment that "Cladding with aluminum can also do the same, so such a statement is not adding new science, or even impact" is too harsh. The excited CO₂ can react with MgO or Mg(OH)₂ at room temperature is one new science we want to demonstrate, and making use of this point to improve the corrosion resistance of Mg alloys is also one of our novelties.

Cladding with aluminum would not work in many applications. For example, the cardiovascular stents or other medical bio-degradable Mg alloy implants cannot be cladded with Al, which is a detrimental element to human bodies, but the MgCO₃ film leaves behind no toxic substance, and can be made sufficiently thin to ensure the elastic compliance and strength of the stents while offering adequate corrosion protection. Also, aluminum cladding is not suitable for micro or nano scaled Mg

alloys devices. Our method has many distinct advantages compared with cladding of Mg alloys with aluminum, such as no series of surface pretreatment, no restriction on the geometry of products, no extra heating, no galvanic corrosion between the coating and Mg, and good adhesion with substrate metal etc.

Comment 7:

The reactions showing Mg going to $MgCO_3$ cannot be accurate, as there is a lot of $Mg(OH)_2$ on the surface. As such, what happens to the $Mg(OH)_2$? is it gone?

Reply:

Firstly, according to our own understanding, the reviewer intended to say that “the reactions showing MgO going to $MgCO_3$ cannot be accurate, as there is a lot of $Mg(OH)_2$ on the surface.....”. And this is a good question.

Our *in situ* experimental results show that the excited CO_2 can react with MgO at room temperature (usually, the reaction of MgO with CO_2 to produce $MgCO_3$ needs a temperature of at least $\sim 400^\circ C$, 1 atm). In the absence of vapor or aqueous phases, the reaction $Mg(OH)_2(s) + CO_2(g) = MgCO_3(s) + H_2O(g)$ also only occurs at high temperature (Fricker, Kyle J., and Ah-Hyung Alissa Park. "Effect of H_2O on Mg $(OH)_2$ carbonation pathways for combined CO_2 capture and storage." *Chemical Engineering Science*, 2013). Similarly, it is reasonable to expect that the excited CO_2 can also react with $Mg(OH)_2$ without the extra heating or the presence of H_2O . Indeed, our real-time observation in E-TEM has proved this point: supplementary Movie 3 shows the rapid reaction of excited CO_2 with the flurry corrosion products produced during the immersion of Mg alloy in water. Even if the corrosion product in water is mainly $Mg(OH)_2$ and doesn't change inside TEM, the by-product vapor can be also evacuated out of the TEM chamber. **Last but not the least, Nordlien *et al.* also reported the water evaporation from the hydrated zones of films formed on Mg and subsequent crystallization of the dehydrated zones into MgO occurring in TEM analysis due to high energy electron irradiation.** (Nordlien, Jan Halvor, *et al.* "Morphology and structure of oxide films formed on MgAl alloys by exposure to air and water." *Journal of the Electrochemical Society*, 1996. Nordlien, J. H., *et al.* "A TEM investigation of naturally formed oxide films on

pure magnesium." Corrosion science ,1997). **So, the carbonation of the air-formed film on our pillars' surface we observed in TEM should be dominated by the reaction $\text{MgO} + \text{CO}_2 = \text{MgCO}_3$.** However, we do agree that the reaction of $\text{Mg}(\text{OH})_2$ with excited CO_2 should not be ignored completely, and relevant parts in the manuscript and supporting information have been modified.

Comment 8:

The list of references is particularly bad Im afraid. There have been excellent Mg surface film observations by Kish et al (all missed!).

Reply:

Kish *et al.* have indeed done a lot of work to study the film on Mg and Mg alloys in aqueous environment. But Kish *et al.*'s cross-sectional samples for TEM examinations were all fabricated using focused ion beam (FIB). It is known that high energy ion bombardment can cause irradiation damage (physical sputtering, temperature increase etc.) of ionic solids, for example MgO and $\text{Mg}(\text{OH})_2$. (Matzke, Hj, and J. L. Whitton. "Ion-bombardment-induced radiation damage in some ceramics and ionic crystals: determined by electron diffraction and gas release measurements." *Canadian Journal of Physics* ,1966). We also tried to use focused ion beam (30 keV Ga^+) to lift out the cross-sectional sample from the bulk Mg immersed in distilled water for 24 hours. We found that the plate-like structure on the surface was very sensitive to ion bombardment even with very low beam current. We clearly observed the morphology change of the surface film during FIB milling. Therefore, we are inclined to trust the results from samples prepared by ultramicrotomy and we cited highly-cited similar works from Nordlien *et al.* (Nordlien, J. H., *et al.* "A TEM investigation of naturally formed oxide films on pure magnesium." *Corrosion science* ,1997; Nordlien JH, Ono S, Masuko N, Nisancioglu K, Morphology and structure of oxide-films formed on magnesium by exposure to air and water. *J Electrochem Soc*, 1995). These papers show obvious discrepancies in the Mg film structures with what reported by Kish *et al.* in "Analysis of the surface film formed on Mg by exposure to water using a FIB

cross-section and STEM-EDS." This should be attributed to the different sample preparation methods.

We can add some references from Kish et al., if the reviewer still feels they are necessary, after seeing our concerns above.

Comment 9:

All that Figure 3b shows is a galvanic couple, not protection at all. One side is the cathode, one the anode. This is how corrosion works..... All it takes is a small potential difference and this occurs. This is not so whopping.

Reply:

We disagree with the reviewer that Figure 3b is only a galvanic couple instead of the protection effect from MgCO₃ film.

Figure 3a has obviously shown that the Mg pillar treated in excited CO₂ is well protected compared with its counterpart without any treatment. And in Figure 3a case, there exists no “galvanic couples” because the entire Mg pillar was coated with MgCO₃. Figure 3b aims to further prove the protection of MgCO₃ film using the same Mg alloy pillar to exclude any possible discrepancies between different samples. Increased corrosion potential of the treated part unambiguously shows the protective effect of MgCO₃ coating, and this increase is significant, as shown in our supplementary electrochemical tests on bulk Mg samples (see Figure S12 and Table S1 in Supporting Information); meanwhile the corrosion current density also decreases by two orders of magnitude. So, the MgCO₃ film formed in excited CO₂ can greatly decelerate the corrosion rate and obviously improve the corrosion resistance of Mg alloys.

Comment 10:

The claim in Figure 4 that "Turning corroded Mg alloy surface into protective coating." is an overstatement. There is quite a lot of metal loss.

Reply:

Figure 4 compares the corrosion resistance of the native oxide/hydroxide layer and MgCO₃ film on the same Mg alloy pillar. We deliberately pre-corroded an Mg alloy pillar and meanwhile made sure that one part of the pillar was uncorroded by tuning the immersion time. And then the corroded part was exposed to e-beam illumination and the flurry corrosion product reacted with excited CO₂ to produce MgCO₃ film wrapping this part (see supplementary movie 3). At that time, the uncorroded top part was bare. After the immersion in water and subsequent exposure in humid air, the originally corroded lower part kept intact with the protection of MgCO₃, while the unprotected top part was seriously corroded, and that caused the so-called “metal loss”.

Therefore, the summary of Figure 4 “Turning corroded Mg alloy surface into protective coating” is drawn from the real and solid experimental evidence.

Comment 11:

Irrespective of much of the above, the authors describe a coating... in essence. It is not a protective coating of value unless it is protective in a defect, or self forming (like a passive film). The Mg-Li film previously demonstrated on Mg-Li alloys is self healing (by self reforming), so it protects defect. Coating Mg with paint or plating with Cu or Ni is common. None of the latter protect defects, and neither does the coating described here. This means it is not actually interesting, or useful.

Reply:

We agree that a self-reforming or self-healing protective film like the passive oxide on the surface of aluminum or titanium would be great. But there are applications where other surface treatments would be desirable, as long as the corrosion rate can be judiciously controlled. And every method has its own advantages and shortcomings. For example, the common chemical conversion coatings with self-healing ability face the toxicity, environmental contamination and poor mechanical properties, etc. Even the “stainless” Mg-Li alloy reported by Xu *et al.* is not always perfect because it is not suitable as the bio-degradable material. The surface treatment method we reported here is not perfect either, but it is easy to

operate, environmentally-benign and especially suitable for pretreating small sized devices and bio-medical implants.

Nowadays, most of Mg alloys are still facing serious poor corrosion resistance problem. Therefore, to develop new corrosion prevention methods is not only necessary but also in a press need. Each Mg alloy used in different applications should has its own specific and optimized corrosion prevention method.

Response to the reviewer #2:

Comment 1:

This is an interesting paper about the use of room temperature pretreatments to improve the protective oxide of Mg. These authors demonstrate via in situ TEM (environmental and mechanical) that the native oxide on Mg can be dramatically improved by exposure to ionizing CO₂ gas (using electron beam) to change the native oxide to a protective MgCO₃ layer. The authors claim that this oxide is more resistant to corrosion and more adherent to the underlying Mg. The work is of high quality, especially the in situ microscopy. Sufficient details in the paper and the supplement are given for others to reproduce the data.

Reply:

We thank the reviewer for pointing out the significance of our work! And we are glad to hear that the reviewer considers our work to be of high quality. We also highly appreciate the valuable suggestions given by the reviewer. In the following, we respond point-to-point to each comment.

Comment 2:

However, the corrosion experiments were done by only immersion in deionized water. John T. Yates, Jr. paper in Langmuir 2001, 17, 2146-2152 also shows that using different oxidation conditions can improve corrosion behavior as well, where they

used an electrochemical cell to verify improved corrosion behavior. This paper could be used as an example of how to further characterize the MgO versus the MgCO₃. The authors could also demonstrate improved oxidation behavior as well, as adhesion of the oxide is important for high temperature corrosion field, where cyclic oxidation using TGA is a standard method to demonstrate improved oxidation behavior.

Reply:

We thank the reviewer for the valuable suggestions. Our original goal of the present work was to explore an environment friendly and efficient method to improve the corrosion resistance of Mg alloys especially at small scale. So, we focused on the method development and proof processes with the convincing in-situ experimental evidences. But it is necessary to further verify the improved corrosion resistance using standard electrochemical corrosion tests. The anti-corrosion properties in 3.5% NaCl solution of the MgCO₃ protective films on the bulk Mg samples were evaluated by using potentiodynamic polarization measurements.

Considering that the native film on Mg surface is only tens of nanometers, the formed MgCO₃ protective layer is too thin to protect the bulk Mg. Inspired by the reaction of excited CO₂ with corrosion product on Mg micropillars' surface, we intentionally immersed the bulk Mg samples in deionized water for some time to form a fluffy pre-corroded layer with the micron-scale thickness.

Firstly, we immersed the bulk pure Mg (99.95 wt.%) samples in deionized water for 24 hours at room temperature to pre-corrode the surface for obtaining Mg(OH)₂ & MgO films. SEM image in Figure S12b shows the plate-like morphology of the film surface formed on Mg sample after immersion. This film was found to contain a top layer of crystalline MgO embedded in an amorphous Mg(OH)₂ matrix with an overall plate-like morphology. (Nordlien, J. H., et al. "A TEM investigation of naturally formed oxide films on pure magnesium." *Corrosion science*, 1997).

These bulk Mg samples with pre-corrosion surface film were treated in CO₂ plasma for 1 hour or 2 hours. The corroded surface with plate-like morphology were carbonated resulting in a relatively flat and dense MgCO₃ surface (see below Figure S12). And then the anti-corrosion properties of MgCO₃ films were evaluated in 3.5

wt.% NaCl solution by using potentiodynamic polarization measurements. Compared with their counterparts, the corrosion resistance of bulk Mg samples with MgCO₃ protective film has been enhanced dramatically: the corrosion potential ($E_{\text{corr vs.SCE}}$) increases, and the corrosion current density (i_{corr}) can decrease by two orders of magnitude (see below Table S1).

After corrosion tests, large corrosion craters can be clearly observed even by naked eyes on pristine Mg surface. But the surfaces of the samples treated in CO₂ for 2 hours are still intact and there are no visible cracks (Figure S12). The anti-corrosion effects of MgCO₃ protective film can be better than the anodized coating, and even can be comparable with the composite coatings obtained by micro-arc oxidations, see Table S1. (Xue, Dingchuan, et al. "Corrosion protection of biodegradable magnesium implants using anodization." *Materials Science and Engineering*, 2011; Zhao, Lichen, et al. "Growth characteristics and corrosion resistance of micro-arc oxidation coating on pure magnesium for biomedical applications." *Corrosion Science*, 2010. Mu, Weiyi, and Yong Han. "Characterization and properties of the MgF₂/ZrO₂ composite coatings on magnesium prepared by micro-arc oxidation." *Surface and Coatings Technology*, 2008.)

The optimal parameters, such as the immersion time in water and CO₂ plasma treatment time should be further explored so as to get optimized anti-corrosion properties.

The newly formed MgCO₃ protective film also displays significant oxidation resistances. See our reply to the comment 4 for experimental details and results. Since the present work mainly focuses on the mechanism and microscopic characterization, we have supplemented in-situ high-temperature oxidation experiments on the Mg pillar with MgCO₃ protective film.

As for the further characterizations of MgO/Mg(OH)₂ vs. MgCO₃, such as thermogravimetric analysis (TGA) to compare their thermal stability, nano-indentation or nano-scratching to show the mechanical properties of the film on Mg and so forth, we are going to report them in another work systematically and comprehensively.

Figure S12. Anti-corrosion effects of MgCO₃ protective film on bulk Mg. Typical SEM images of the Mg samples before and after potentiodynamic polarization tests: **(a)** pristine pure Mg sample without any treatments; **(b)** the pure Mg sample immersed in deionized water for 24 h; **(c)** the pure Mg sample immersed in deionized water for 24 h and then treated in CO₂ plasma for 1 h; **(d)** the pure Mg sample immersed in deionized water for 24 h and then treated in CO₂ plasma for 2 h. **(e)** corresponding potentiodynamic polarization curves of above-mentioned Mg samples.

Table S1

The results of potentiodynamic corrosion tests in 3.5 wt.% NaCl solution and comparisons with anodizing and micro-arc treatment.

Samples	E_{cor} (V_{SCE})	i_{cor} ($\mu\text{A}/\text{cm}^2$)
Pristine Mg	-1.58	196.64
Immersion 24h	-1.59	251.28
Immersion 24h+plasma 1h	-1.54	19.52
Immersion 24h+plasma 2h	-1.06	0.31
Immersion 24h+plasma 2h	-0.73	0.22
Anodizing coating	-1.48	27
Composite coating by micro-arc oxidation	-1.49	0.69
Micro-arc oxidation coating	-1.69	0.17

Comment 3:

Another point is that more specifics of the SAED analysis could be given to be more convincing that the oxide formed is MgCO₃ (or mixed with MgO).

Reply:

We thank the reviewer for this good advice. We made the pure nanoscale MgO crystal react with excited CO₂, and by indexing the diffraction pattern of the product we verified the product was MgCO₃. Accordingly, it is reasonable to conclude that the oxide layer on Mg pillar surface reacting with excited CO₂ should be MgCO₃. We also tried to use the EELS analysis (carbon element signal) and SAED to further confirm that the product is MgCO₃ (see Figure S4 in supporting information). As for more specifics of the SAED, it is impossible with our TEM to select the MgCO₃ film only, which is several nanometers thick using the selected area aperture. Therefore, we took the high resolution TEM images of the surface areas of the Mg pillar treated in excited CO₂, and then we could get the diffraction information of very small localized region via fast Fourier transform (FFT). The figure shown below is the HRTEM characterization of the surface layer after treatment in excited CO₂. Very small zone with the size of only several nanometers can be selected to make FFT, which was indexed to be crystalline MgCO₃ by comparing with the standard diffraction pattern. This confirms the existence of MgCO₃.

Figure S5. High-resolution TEM characterization and corresponding FFT of the surface layer after treatment in excited CO₂. (a) Standard diffraction pattern of crystalline Mg with the [001] zone axis (ZA). (b) Standard diffraction pattern of crystalline MgCO₃ with ZA=[001]. (c) Bright field TEM image of the Mg nanopillar after treated in 2 Pa excited CO₂. (d) HRTEM image of the framed zone in (c), and the inset image is the corresponding FFT of the marked zone. The FFT coincides with the diffraction information of MgCO₃ in (b), and this indicates this small area only contains crystalline MgCO₃. (e) HRTEM image of the framed zone in (c), and (f) is the corresponding FFT of the marked zone. The selected area is larger, so more information in reciprocal space can be seen: the spots circled in yellow come from monocrystalline Mg substrate, and the spots circled in blue derive from crystalline

MgCO₃; other unmarked dispersed spots should represent the unreacted and crystallized oxide.

Action taken:

We have included this part in supporting information as Figure S5. And the figure indices in SI have been updated accordingly.

Comment 4:

As the authors have access to an ETEM, could the authors expose the samples *in situ* to O₂ or H₂O vapor to determine if the MgCO₃ in contrast to the initial MgO does slow down or change oxidation behavior at higher temperatures?

Reply:

We appreciate this good suggestion and carried out the *in situ* heating tests in gas environment. Because water vapor is not one of the standard gases recommended by the Hitachi environmental TEM, we exposed the samples in O₂ at high temperature to observe the different responses of the Mg pillars with and without the MgCO₃ coating. The two figures below were extracted from the *in situ* videos recorded during the heating processes (from 20 °C to 200 °C) of the Mg pillars with native oxide layer and MgCO₃ layer, respectively. Clearly, we can see that the native oxide layer couldn't protect the substrate metal from the oxygen attack at high temperature and the Mg pillar was seriously oxidized. In contrast, the morphology of Mg pillar with MgCO₃ protective film almost kept unchanged and the film was also intact to some extent at 200 °C, which proves the protective effect of the contact MgCO₃ film once again unambiguously. Presumably, this is because the compact and stable MgCO₃ film can effectively isolate the oxygen gas from Mg metal for the tested temperature range.

It should be noted that reaching 200 °C is a quite harsh environment for the nanoscale pillars.

Figure S8. Comparison of the high temperature oxidation of the Mg pillars with and without MgCO₃ protective film. (a) Real-time oxidation process of as-FIBed pure Mg nanopillar with the increasing temperature from 20□ to 200□ in 2 Pa O₂ gas. **(b)** Real-time heating process of the Mg nanopillar with MgCO₃ protective film with the increasing temperature from 20□ to 200□ in 2 Pa O₂.

Action taken:

We have included this part in supporting information as Figure S8.

Comment 5:

In the conclusion section, it would also be beneficial to the general audience to reiterate the broad application of Mg in technology and what environments that Mg would be exposed to in these technological applications. It would also be helpful to again provide some thermodynamics or kinetics of this reaction ($\text{MgO} + \text{CO}_2 \rightarrow \text{MgCO}_3$) in the conclusion section in order to give a broader implication of how to select the pre-treatments to create better protective oxides on other metallurgical systems.

Reply:

Thanks a lot for the reviewer's advices. In the conclusion section, we have reiterated the broad applications and exposure environment of Mg alloys in service, and also the thermodynamics of the carbonation reaction.

Action taken:

In the conclusion section, we have modified the part before "The entire surface treatment process has been recorded in real time....." as "In summary, Mg alloys have broad uses in 3C products, automotive, aerospace and biomedical industries. However, all of these applications face corrosion resistance problem in service, especially in the humid or aqueous environments. We developed an easy, environment-benign and effective anti-corrosion method: carbonation of the air-formed oxide/hydroxide film or hydrate corrosion products on Mg alloys' surface into a smooth, compact MgCO_3 protective layer by ionizing CO_2 gas using either high energy electron beam or plasma. The excited CO_2 accomplishes at room temperature the reaction $\text{MgO} + \text{CO}_2 \rightarrow \text{MgCO}_3$, which usually occurs above 400°C at atmospheric pressure. No extra heating or pretreatments are needed, rendering this method especially suitable for protecting small-sized or complex-shaped Mg alloy workpieces and for replenishing those already-corroded. Rather than having to mechanically clean away prior corrosion damage, one can consume it directly to create new protective surface."

Comment 6:

Some minor points:

SM video 2: the beginning should be edited out.

Subtitles should be removed (e.g., Verification of Idea).

Some careful minor edits would improve clarity (e.g., Figure 2b"... MgO flakes kept unchanged..." would read better as"... MgO flakes do not change...")...

Reply:

We have made corrections accordingly in the manuscript and supplementary materials.

Action taken:

The beginning part with e-beam blocked off in movie 2 has been edited out.

Subtitles in manuscript have been removed.

Caption in Figure 2b has been rewritten as "...MgO flakes did not change...."

Comment 7:

To summarize, this is an interesting paper that should influence the field of corrosion, oxidation, and nanoscience and technology, as the authors clearly demonstrate that different pretreatments especially reactive gas species can improve properties significantly. Although the TEM is excellent and convincing, the claim that the corrosion properties are improved through exposure to deionized water could be better. It would benefit the paper to carry out the more standard electrochemical corrosion or higher temperature cyclic oxidation tests to demonstrate that this pretreatment improves Mg resistance to corrosion or oxidation properties.

Reply:

We thank the reviewer for this concise summary of our work. We also really appreciate these valuable suggestions.

We have modified our conclusion as “the corrosion resistance of Mg alloys in deionized water can be improved”. And as shown above, we have supplemented the standard electrochemical corrosion tests and higher temperature oxidation experiments to prove the superior anti-corrosion properties and stability of the MgCO₃ protective film.

Response to the reviewer #3:

Comment 1:

This paper demonstrated that magnesium oxide could be converted into magnesium carbonate by excited CO₂ in an environmental TEM, and the formed magnesium carbonate film could enhance the mechanical properties and inhibit the corrosion of magnesium in water. These are innovative findings. The most impressive experiment in this study was the in-situ observation of the formation process of magnesium carbonate. Therefore, the reviewer would like to recommend publication of this paper.

Reply:

Thank the reviewer for the accurate and comprehensive summary of our work. We are glad to hear that the reviewer considers our work to be scientifically sound and to be of general interests to the readers of *Nature Communications*.

Comment 2:

However, this paper can be further improved if the following issue could be properly addressed.

Corrosion resistance is critical for a surface film on magnesium. Since the formed magnesium carbonate film in this paper was claimed to have superior corrosion resistance in water, it would be worthwhile to compare this film with a simple coating, e.g. an anodized coating, on the magnesium under the same condition. The reviewer believes that such a comparison will interest many more scientists and engineers. If the magnesium carbonate film can prove to be more corrosion resistant than the simple coating, then this new film formation technique may open up a new way of coating magnesium alloys in practical applications.

Reply:

We appreciate very much this valuable suggestion made by the reviewer. The anti-corrosion properties of the MgCO_3 protective films have been evaluated in 3.5 wt.% NaCl solution by using potentiodynamic polarization measurements and the results, including comparisons with other types of simple coatings, have now been included in supporting information.

Considering that the native film on Mg surface is only tens of nanometers, the formed MgCO_3 protective layer is too thin to protect the bulk Mg. Inspired by the reaction of excited CO_2 with corrosion product on Mg micropillars' surface, we intentionally immersed the bulk Mg samples in deionized water for some time to form a plate-like pre-corroded layer with the micron-scale thickness.

Firstly, we immersed the bulk pure Mg (99.95 wt.%) samples in deionized water for 24 hours at room temperature to pre-corrode the surface for obtaining $\text{Mg}(\text{OH})_2$ & MgO films. SEM image in Figure S12b shows the plate-like morphology of the film surface formed on Mg sample after immersion. This film was found to contain a top layer of crystalline MgO embedded in an amorphous $\text{Mg}(\text{OH})_2$ matrix with an overall plate-like morphology. (Nordlien, J. H., et al. "A TEM investigation of naturally formed oxide films on pure magnesium." *Corrosion science*, 1997).

These bulk Mg samples with pre-corrosion surface film were treated in CO_2 plasma

for 1 hour or 2 hours. The corroded surface with plate-like morphology were carbonated resulting in a relatively flat and dense MgCO_3 surface (see below Figure S12). And then the anti-corrosion properties of MgCO_3 films were evaluated in 3.5 wt.% NaCl solution by using potentiodynamic polarization measurements. Compared with their counterparts, the corrosion resistance of bulk Mg samples with MgCO_3 protective film have been enhanced dramatically: the corrosion potential ($E_{\text{corr vs. SCE}}$) increases, and the corrosion current density (i_{corr}) can decrease by two orders of magnitude (see below Table S1). After corrosion tests, large corrosion craters can be clearly observed even by naked eyes on pristine Mg surface, but the surfaces of the samples treated in CO_2 for 2 hours are still intact and there are no visible cracks (Figure S12).

Note that the anti-corrosion effects of MgCO_3 protective film are much better than the anodized coating, and even can be comparable with the composite coatings obtained by micro-arc oxidations, see Table S1. (Xue, Dingchuan, et al. "Corrosion protection of biodegradable magnesium implants using anodization." *Materials Science and Engineering*, 2011; Zhao, Lichen, et al. "Growth characteristics and corrosion resistance of micro-arc oxidation coating on pure magnesium for biomedical applications." *Corrosion Science*, 2010. Mu, Weiyi, and Yong Han. "Characterization and properties of the MgF2/ZrO2 composite coatings on magnesium prepared by micro-arc oxidation." *Surface and Coatings Technology*, 2008.)

Therefore, we can conclude that the magnesium carbonate film can prove to be more corrosion resistant than the simple coating, and as the reviewer said that this new film formation technique may open up a new way of coating magnesium alloys in practical applications. But the optimal parameters, such as the immersion time in water and CO_2 plasma treatment time should be further explored so as to get more remarkable anti-corrosion properties.

Figure S12. Anti-corrosion effects of MgCO₃ protective film on bulk Mg. Typical SEM images of the Mg samples before and after potentiodynamic polarization tests: **(a)** pristine pure Mg sample without any treatments; **(b)** the pure Mg sample immersed in deionized water for 24 h; **(c)** the pure Mg sample immersed in deionized water for 24 h and then treated in CO₂ plasma for 1 h; **(d)** the pure Mg sample immersed in deionized water for 24 h and then treated in CO₂ plasma for 2 h. **(e)** corresponding potentiodynamic polarization curves of above-mentioned Mg samples.

Table S1

The results of potentiodynamic corrosion tests in 3.5 wt.% NaCl solution and comparisons with anodizing and micro-arc treatment

Samples	E_{cor} (V_{SCE})	i_{cor} ($\mu\text{A}/\text{cm}^2$)
Pristine Mg	-1.58	196.64
Immersion 24h	-1.59	251.28
Immersion 24h+plasma 1h	-1.54	19.52
Immersion 24h+plasma 2h	-1.06	0.31
Immersion 24h+plasma 2h	-0.73	0.22
Anodizing coating	-1.48	27
Composite coating by micro-arc oxidation	-1.49	0.69
Micro-arc oxidation coating	-1.69	0.17

Reviewers' comments:

Reviewer #1 (Remarks to the Author):

The revision is worse than the original, as the authors are oblivious to the shortcomings. Right from the title, the paper is focused on corrosion resistant coating. As such, it can only be viewed in that context (and not viewed in the context of what the paper isn't). In terms of corrosion resistance, the Mg-carbonate is not self-healing, it is likely to be soluble when wet, and it is not anywhere near as novel as the authors claim.

All that figure 3B is showing is a galvanic couple, not corrosion protection.

This really is a very poor paper.

Once again, the authors have avoided the might of the literature and opted to cite irrelevant textbooks in oxidation, etc.

This paper would not survive pre-screening at a corrosion journal, and I cannot in good faith say it is anywhere near suited to Nature Comms.

My recommendation is a firm reject.

Reviewer #3 (Remarks to the Author):

This reviewer is happy with the revision.

Reviewer #4 (Remarks to the Author):

The submitted manuscript introduces a detailed experimental study showing that the native surface oxide on Mg can be converted MgCO₃ via e-beam irradiation (or plasma) in a CO₂ atmosphere. With a set of corrosion tests, the authors further show that the resulting MgCO₃ surface layer has improved corrosion resistance compared to the native surface oxide. These results are interesting and the experimental evidence is convincing. However, the following issues including the fundamental mechanism regarding the e-beam assisted MgCO₃ formation require careful clarification:

1) The e-beam assisted MgCO₃ formation is attributed to the excitation of CO₂ gas molecules. This point can be incorrect and is against the experimental results. As shown in Figs. 3 and 4, the MgCO₃ formation happens only in the local area under the e-beam irradiation, suggesting clearly that only adsorbed CO₂ species are activated by the e-beam. In other words, the e-beam ionizes the CO₂ molecules that have already adsorbed on the surface instead of CO₂ molecules in the gas phase. Otherwise, MgCO₃ formation would not be limited only to the e-beam irradiation area due to the fast and random motion of gas molecules. This effect can be similar to the electron-bombardment effect on promoting the surface oxidation of Al(111), as shown by John Yates (PRL 89, 276101 (2002)). In this sense, the authors should consider carefully the fundamental difference between e-beam and plasma in their experiments, because the latter may excite CO₂ molecules in the gas phase, provided that the MgCO₃ formation also happens in hidden areas (not directly exposed to the plasma).

2) Fig. 5 shows that the CO₂ treated Mg pillar has improved mechanical properties compared to the untreated one. This is reasonable because the MgCO₃ layer examined in their in-situ TEM experiment is very thin. The reviewer agrees with the authors' statement "there should be a critical thickness above which the MgCO₃ scale would behave similarly to its brittle bulk counterpart". This also points to a question about the necessities of controlling the plasma-assisted carburization of the corroded product to avoid mechanical failure for thick MgCO₃. The authors may add some comments.

3) There are some minor points that need to fix such as:

i) the definition of the PB ratio is incorrect. It should be the ratio of the molar volume of the oxide with the molar volume of the metal.

ii) the statement "the waterproof carbonate can interfere with both anodic and cathodic reaction ..." is vague and confusing. Does the surface layer chemically react or non-react with water? Improved clarity is needed.

4) there are some English errors that require careful proof read, such as the missing of "the" at many places.

5) this reviewer also went through the review comments and the authors' response from the first round of review. Most of the comments from reviewer 1 appear focusing on the practical aspect of the work reported in this manuscript, which I feel are minor issues in view of the fundamental implication of this study.

Response to the reviewer #1:

Comment 1:

The revision is worse than the original, as the authors are oblivious to the shortcomings. Right from the title, the paper is focused on corrosion resistant coating. As such, it can only be viewed in that context (and not viewed in the context of what the paper isn't).

This really is a very poor paper..... This paper would not survive pre-screening at a corrosion journal, and I cannot in good faith say it is anywhere near suited to Nature Comms. My recommendation is a firm reject.

Reply:

We feel sorry that we did not convince the reviewer in our previous reply. However, the comments from other reviewers clearly contradict the judgement from referee 1. For example, they remarked that *“The work is of high quality, especially the in situ microscopy. Sufficient details in the paper and the supplement are given for others to reproduce the data.”*, *“this is an interesting paper that should influence the field of corrosion, oxidation, and nanoscience and technology, as the authors clearly demonstrate that different pretreatments especially reactive gas species can improve properties significantly”*, *“These are innovative findings. The most impressive experiment in this study was the in-situ observation of the formation process of magnesium carbonate. Therefore, the reviewer would like to recommend publication of this paper”*, *“These results are interesting, and the experimental evidence is convincing.”*, and *“Most of the comments from reviewer 1 appear focusing on the practical aspect of the work reported in this manuscript, which I feel are **minor issues** in view of the fundamental implication of this study.”*

In the following, we address the concerns from reviewer 1 in detail.

Comment 2:

Once again, the authors have avoided the might of the literature and opted to cite irrelevant textbooks in oxidation, etc.

Reply:

Based on the reviewer's comments from last round, we guess "the might of the literature" here refers to Esmaily *et al.*'s review paper (*2017 Progress in Materials Science*) and Kish *et al.*'s papers about the observations of Mg surface film, and the "irrelevant textbooks in oxidation" refers to Nordlien *et al.*'s works, as we described in our last response.

Action taken: according to the reviewer's suggestion, we have cited Esmaily *et al.*'s review paper (Ref.18 on page 3 as "it was reported that the protective carbonates/hydroxy carbonates-containing layer on the surface of Mg alloys can suppress corrosion by blocking the anodic and/or cathodic sites^{17,18}") in the introduction part to demonstrate the inhibitive effect of Mg carbonate/hydroxy-carbonate, which prompted us to find a novel method to grow the compact MgCO₃ protective film at room temperature.

Further explanation: we'd like to explain why we cited the relevant oxidation work from Nordlien *et al.* instead of Kish's similar works. Kish *et al.* have indeed done a lot of work to study the film on Mg and Mg alloys in aqueous environment. But Kish *et al.*'s cross-sectional samples for TEM examinations were all fabricated using focused ion beam (FIB). It is known that high energy ion bombardment can cause irradiation damage (physical sputtering, temperature increase etc.) of ionic solids, such as MgO and Mg(OH)₂. We also tried to use focused ion beam (30 keV Ga⁺) to lift out the cross-sectional sample from the bulk Mg immersed in distilled water for 24 hours. We found that the plate-like structure on the surface was very sensitive to ion bombardment even with very low beam current and clearly observed the morphology change of the surface film during FIB milling. While Nordlien's Mg samples were fabricated by ultramicrotomy, which is FIB-free and more convincing to demonstrate

the naturally formed oxide layers on Mg. Therefore, we cited those highly-cited relevant works from Nordlien *et al.*

Comment 3:

In terms of corrosion resistance, the Mg-carbonate is not self-healing, it is likely to be soluble when wet, and it is not anywhere near as novel as the authors claim.

Reply:

We agree that Mg-carbonate is not self-healing. However, this does not preclude it as a useful surface protection. The MgCO_3 solubility in water at room temperature is as low as 0.02%. This explains why MgCO_3 can exist in nature for years. In addition, one proposed application of our finding is to manufacture bio-degradable Mg alloys with controllable corrosion rate. This means that even though the materials are not self-healing, they can be still very useful in certain circumstances if they are designed and fabricated in an appropriate manner.

Our work has significant novelty, which has been resonated by many experts in the field, including other reviewers. Firstly, our work is based on an in-depth understanding as to how MgO reacts with activated CO_2 . The applications and effects we showed met our design expectations. Even though previous work indicated that magnesium hydroxy carbonate or carbonate produced in humid or aqueous environment have certain anti-corrosion effect, the effects are usually weak, and the working principles are buried under complex conditions. Secondly, almost all Mg alloys in the atmospheric environment containing CO_2 are still facing serious corrosion problem, that's why so many researchers keep looking for different methods to improve their corrosion resistance. Our novelty mainly lies in understanding the nature of the reaction between the activated CO_2 and Mg oxide or hydroxide at room temperature and then apply it purposely. The environmental TEM technique, which is rarely used in the corrosion community, played a key role in inspiring our findings.

Comment 4:

All that figure 3B is showing is a galvanic couple, not corrosion protection.

Reply:

We cannot agree with the reviewer that Figure 3b is only a galvanic couple instead of the protection effect from MgCO_3 film. Firstly, the corrosion potential of the protected top part is indeed higher than the unprotected bottom part, which can be concluded from our potentiodynamic polarization tests results. So, these two parts with different corrosion potentials form a non-typical galvanic couple, which is usually developed when two different metals are separated by electrolytes. The increased corrosion potential of the top part with MgCO_3 film unambiguously vindicates our point that the produced MgCO_3 film has obvious protective effect.

Response to the reviewer #4:

Comment 1:

The submitted manuscript introduces a detailed experimental study showing that the native surface oxide on Mg can be converted MgCO_3 via e-beam irradiation (or plasma) in a CO_2 atmosphere. With a set of corrosion tests, the authors further show that the resulting MgCO_3 surface layer has improved corrosion resistance compared to the native surface oxide. These results are interesting and the experimental evidence is convincing.

Reply:

We thank the reviewer for pointing out the significance of our work! And we are glad to hear that the reviewer considers our work to be convincing. We also highly appreciate the valuable suggestions given by the reviewer. In the following, we respond point-to-point to each comment.

Comment 2:

However, the following issues including the fundamental mechanism regarding the e-beam assisted MgCO_3 formation require careful clarification:

The e-beam assisted MgCO_3 formation is attributed to the excitation of CO_2 gas

molecules. This point can be incorrect and is against the experimental results. As shown in Figs. 3 and 4, the MgCO_3 formation happens only in the local area under the e-beam irradiation, suggesting clearly that only adsorbed CO_2 species are activated by the e-beam. In other words, the e-beam ionizes the CO_2 molecules that have already adsorbed on the surface instead of CO_2 molecules in the gas phase. Otherwise, MgCO_3 formation would not be limited only to the e-beam irradiation area due to the fast and random motion of gas molecules. This effect can be similar to the electron-bombardment effect on promoting the surface oxidation of Al (111), as shown by John Yates (PRL 89, 276101 (2002)). In this sense, the authors should consider carefully the fundamental difference between e-beam and plasma in their experiments because the latter may excite CO_2 molecules in the gas phase, provided that the MgCO_3 formation also happens in hidden areas (not directly exposed to the plasma).

Reply:

We really appreciate the constructive comments and excellent advices from the reviewer. We fully agree that it is the adsorbed CO_2 species ionized by the electron beam that play the key role in forming the MgCO_3 layer. The Yates' paper (PRL 89, 276101, 2002) mentioned by the reviewer demonstrated that electron bombardments can enhance the oxidation rate of Al by inducing the negative electrostatic potential stored on the outer oxide film. In addition, the electrostatic field results in "memory effect" for prior electron irradiation. In other words, the enhanced oxidation continues even after e-beam bombardment is stopped. In our case, the carbonation discontinued as soon as the e-beam was blocked off, and it restarted immediately after the e-beam was turned on again (see below Figure S3), which suggests that the reaction of MgO with excited CO_2 species has no obvious memory for prior electron irradiation at all. Therefore, the reaction of MgO with CO_2 at room temperature should be attributed mainly to the electron beam-stimulated excitation of adsorbed CO_2 species on the oxide surface. At the same, e-beam irradiation-induced defect sites or the electrostatic field effect within the oxide film mentioned in Yates's work, even if existed, should not play any significant role in our case.

Based on aforementioned consideration, we added the following discussions to compare the different effects of electron beam and plasma on the formation mechanism of MgCO_3 film: “It should be noted that the reaction mechanism of MgO with e-beam excited CO_2 is fundamentally different with CO_2 plasma. The glow discharge activated CO_2 molecules in the gas phase directly form the CO_2 plasma with ionic fragments and radicals of CO_2^* . Due to the fast motion of gas species, the formation of MgCO_3 occurs in all the areas that can be reached by the CO_2 plasma. However, the situation is different for high energy e-beam irradiation. Compared to the CO_2 species adsorbed on the native oxide surface, the ionization rate of those free moving CO_2 gas molecules is much lower²³. Therefore, MgCO_3 forms only in the local areas with adsorbed CO_2 species and being exposed to the e-beam irradiation.”

Actions taken:

1> We have added a new figure in supporting information as Figure S3 to show that the reaction of MgO with excited CO_2 species has no obvious memory effect for the prior e-beam irradiation.

Figure S3. The reaction of MgO with excited CO_2^* shows no obvious “memory” effect for the prior electron beam irradiation. (a) The TEM bright field image of pristine MgO flakes. **(b)** The morphology of products after exposure in 2 Pa CO_2 with the e-beam irradiation for 8 minutes. **(c)** The products kept unchanged in 2 Pa CO_2 with the e-beam off for 6 minutes, indicating that prior e-beam irradiation effect disappears as soon as e-beam is discontinued. The reaction restarted again as soon as the e-beam was turned on. **(d)** The morphology of products after exposure in 2 Pa CO_2 with the e-beam irradiation for additional 6 minutes.

2> We modified the right-side schematic in Figure 1 to demonstrate that the excited CO_2^* species mainly distribute on the oxide surface rather than suspend in the gas environment.

Figure 1 in the revised manuscript

3> All statements related to the formation mechanism of the MgCO_3 film have been modified in the manuscript. Those revised parts are highlighted in blue on page 4, 5, 6, and 8, respectively. We also added discussions to compare the different effects of electron beam and plasma on the formation mechanism of MgCO_3 film (see page 10).

Comment 3:

Fig. 5 shows that the CO_2 treated Mg pillar has improved mechanical properties compared to the untreated one. This is reasonable because the MgCO_3 layer examined in their in-situ TEM experiment is very thin. The reviewer agrees with the authors' statement "there should be a critical thickness above which the MgCO_3 scale would behave similarly to its brittle bulk counterpart". This also points to a question about the necessities of controlling the plasma-assisted carburization of the corroded product to avoid mechanical failure for thick MgCO_3 . The authors may add some comments.

Reply:

We thank the reviewer for the comment and suggestion.

Action taken:

We have added following comments on the discussion part of the revised manuscript (page 11). “Therefore, for bulk applications, the MgCO_3 layer should be optimized so that it is thick enough to maximize the corrosion inhibition effect while at the same time thin enough to ensure its deformation compatibility. This can be achieved by optimizing the treatment parameters used to generate the MgCO_3 layer.”

Comment 4:

There are some minor points that need to fix such as:

- i) the definition of the PB ratio is incorrect. It should be the ratio of the molar volume of the oxide with the molar volume of the metal.
- ii) the statement “the waterproof carbonate can interfere with both anodic and cathodic reaction ...” is vague and confusing. Does the surface layer chemically react or non-react with water? Improved clarity is needed.
- iii) There are some English errors that require careful proof read, such as the missing of “the” at many places.

Reply:

We thank the reviewer for pointing out these errors.

Action taken:

- i) We have corrected the definition of PB ratio as “the PB ratio is defined as the ratio of the molar volume of a metal oxide to the molar volume of the corresponding metal” on page 3.
- ii) Usually, Mg carbonate is quite stable with a very low solubility of 0.02% in water at room temperature, and it only reacts with water under heating condition. For clarity, the following modification has been made:
“In this work, MgCO_3 was chosen because it is very stable in nature¹⁶ (e.g. it has an ultra-low solubility of 0.02g/100 mL in water at ambient temperature and pressure, and can remain intact in water for quite a long time) and sufficiently compact (by assuming that MgCO_3 instead of MgO grows on Mg metal surface directly, the PB

ratio of MgCO_3 can be calculated to be 2.04). More importantly, it was reported that the protective carbonates/hydroxy carbonates-containing layer on the surface of Mg alloys can suppress corrosion by blocking the anodic and/or cathodic sites^{17,18}, on page 3.

iii) We looked through the whole manuscript and these syntax errors have been corrected.

REVIEWERS' COMMENTS:

Reviewer #4 (Remarks to the Author):

The authors have addressed all my concerns satisfactorily. The work presented in the manuscript is solid and the results are interesting with a great of detail, I recommend it for publication.

Reviewer #5 (Remarks to the Author):

I have reviewed all of the documentation that was presented in support of the submitted manuscript and the review comments and associated responses. I feel the concerns raised by Reviewer #1 are fair but do not weaken the conclusions made by the authors. I think they do have an interesting and original surface modification approach that shows promise towards improved corrosion protection of Mg alloys. I would ask the authors turn down the definitive claims about improved corrosion protection, particularly as they apply to structural components in the transportation industry until validated by standardized accelerated testing protocols on much larger test samples. As mentioned, this can be simply down by referring to the modification technique as a promising new approach.

Response to the issues raised by referees

Response to the referee #4:

Remarks:

The authors have addressed all my concerns satisfactorily. The work presented in the manuscript is solid and the results are interesting with a great of detail, I recommend it for publication.

Reply:

We truly thank the reviewer for the insightful comments and suggestions which have inspired us to improve the manuscript further and make this work better.

Response to the referee #5:

Comment 1:

I have reviewed all of the documentation that was presented in support of the submitted manuscript and the review comments and associated responses. I feel the concerns raised by Reviewer #1 are fair but do not weaken the conclusions made by the authors. I think they do have an interesting and original surface modification approach that shows promise towards improved corrosion protection of Mg alloys.

Reply:

We appreciate the referee's fair judgement and constructive comments, and we are glad to hear that the reviewer considers our work being promising towards improved corrosion protection of Mg alloys.

Comment 2:

I would ask the authors tune down the definitive claims about improved corrosion protection, particularly as they apply to structural components in the transportation industry until validated by standardized accelerated testing protocols on much larger test samples. As mentioned, this can be simply down by referring to the modification

technique as a promising new approach.

Reply:

We really appreciate the advice from the reviewer. We have toned down all the definitive claims about the improved corrosion protection effects on bulk Mg alloys in the manuscript by using “promising”, “potentially” and “to be expected” etc. instead of “universal” and other definitive statements in our manuscript. We also remove the prospective applications of our approach on the automotive and aerospace industries in the conclusion section.

Actions taken:

In Abstract, we rephrase the sentence as “This environment-friendly surface treatment method is promising to protect Mg alloys, including those already-corroded on the surface.”

In Conclusion section, we remove the statement of “In summary, Mg alloys have broad uses in 3C products, automotive, aerospace and biomedical industries. However, all of these applications are impeded by the poor corrosion resistance of this material in service, especially in the humid or aqueous environments. We developed an easy, environment-benign and effective corrosion inhibition method.”